# MEDICAL DECISION TREE-ENHANCED LLMS FOR INTERPRETABLE REASONING

## ABSTRACT

Large Language Models have made significant strides in medical reasoning. However, challenges remain due to their limited medical knowledge and the risk of hallucinations. While RAG methods can mitigate these issues by retrieving relevant medical information, they typically supply verbose text fragments, which challenges the model's comprehension. Inspired by the widespread use and inherent interpretability of medical decision trees in clinical practice, we propose Medical Decision Tree RAG (MDT-RAG), a novel RAG framework specifically designed for medical reasoning. In this approach, clinical guidelines containing diagnostic and therapeutic information are first converted into decision trees, which are then used to augment LLMs in place of raw text. Experiments demonstrate that our method not only enhances the performance of medical LLMs in reasoning tasks but also exhibits strong interpretability. All related resources have been made publicly available[1].

## 1 INTRODUCTION

Large language models (LLMs) have demonstrated impressive language capabilities (OpenAI, 2023; Bai et al., 2023). However, these models are also plagued by hallucinations (Huang et al., 2025; Ji et al., 2023), which produce information that does not match the facts. In the field of medical reasoning, such as clinical diagnosis and treatment, hallucinations are mainly caused by two factors: first, LLMs lack relevant medical knowledge (Pal et al., 2023), which can be alleviated by retrieval-augmented generation (RAG) methods (Gao et al., 2023; Lewis et al., 2020). By retrieving and contextualizing knowledge related to the question, the model can be assisted in reasoning. The second reason is that clinical diagnosis and treatment represent complex reasoning challenges, requiring LLMs to possess sophisticated reasoning abilities. Otherwise, even with access to relevant medical knowledge, LLMs may struggle to perform correct reasoning (Plaat et al., 2024). To enhance LLMs' reasoning capabilities, common approaches include scaling up model size (Ruan et al., 2024) or employing techniques like Reinforcement Learning for fine-tuning (Team et al., 2025; Guo et al., 2025), both of which demand substantial computational resources and large-scale training data.

In a typical medical reasoning scenario, doctors need to infer what specific disease a patient has and determine the appropriate treatment based on the patient's symptoms and test results. This requires doctors not only to have corresponding medical knowledge but also to possess strong logical analysis skills (Schwartz & Elstein, 2008) to exclude possible interference and ultimately make the right decision. Existing RAG methods typically retrieve verbose text fragments (Amugongo et al., 2025) related to the problem or obtain corresponding entities and relationships through knowledge graph retrieval (Zhao et al., 2025; Peng et al., 2024). This retrieved information is often raw and requires the LLM to perform significant analysis and integration with the patient's specific information before reaching a decision. This places a high demand on the LLM's inherent reasoning capabilities. Moreover, the process of the LLM's analysis and integration is opaque, i.e., a black box, with poor explainability (Zhao et al., 2024).

To enhance the medical reasoning of LLMs, we propose Medical Decision Tree Retrieval-Augmented Generation (MDT-RAG). This approach leverages medical decision trees (MDTs) to augment LLMs. MDTs are tree-structured decision models widely used in Clinical Decision Support Systems (CDSS) (Podgorelec et al., 2002).

---

[1] https://github.com/anonymous-user10/MDT-RAG

In this structure, each internal node represents a clinical decision condition based on features, and each branch corresponds to a possible outcome of that condition, and each leaf node provides a final diagnostic conclusion or treatment recommendation. This structure guides doctors step-by-step from the root to an appropriate leaf node (Zhu et al., 2022). The advantages of MDTs lie in their transparency, explainability, and ability to mirror clinical reasoning patterns. They are widely used in clinical decision guidance and clinician education. By utilizing decision trees, MDT-RAG can directly provide the LLM with structured reasoning guides relevant to the patient's specific conditions. This allows the LLM to follow these predefined guides, reducing the demand on its inherent reasoning capabilities while providing enhanced explainability.

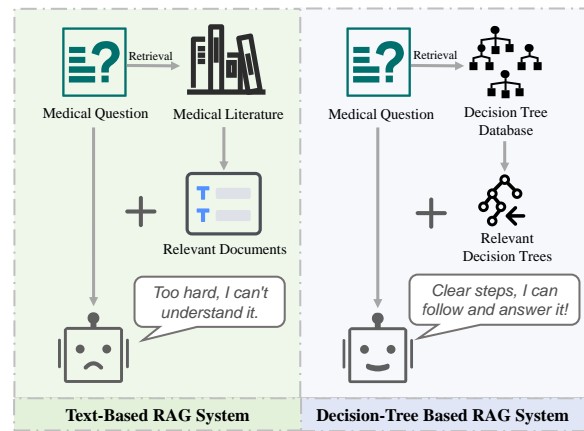

Figure 1: Comparison of Text-Based RAG and MDT-Guided Reasoning

A key challenge in implementing our MDT-RAG is the scarcity of large-scale, open-source MDT datasets. Although they are widely used in CDSS, open-source MDT datasets remain scarce (He et al., 2024). To address this, we built an MDT dataset. We collected clinical guidelines on medical diagnosis and treatment from diverse authoritative sources. Then, we extracted MDTs from the guidelines, which formed our retrieval source. To evaluate the performance of MDT-RAG, we specifically developed a medical reasoning benchmark.

In summary, our contributions are three-fold:

- We propose MDT-RAG, a novel RAG framework that leverages MDTs to assist LLMs in medical reasoning and significantly enhances the interpretability of the reasoning process. To our knowledge, MDT-RAG is the first framework to incorporate MDTs into the RAG system for enhancing clinical reasoning.

- We constructed and open-sourced a large-scale MDT dataset, currently the most extensive publicly available dataset of its kind, along with a medical reasoning benchmark for evaluation.

- Experiments show that MDT-RAG achieves higher performance on medical reasoning tasks while offering enhanced interpretability, a capability particularly valued in clinical decision-making. Our approach provides valuable insights into building more interpretable medical reasoning systems.

## 2 RELATED WORK

**Retrieval-Augmented Generation.** RAG has emerged as a promising paradigm across diverse medical applications, including medical question-answering (Xiong et al., 2024; Wang et al., 2023c; Jeong et al., 2024), text generation (e.g., EHRs, reports) (Zhu et al., 2024b; Wang & Sun, 2022), literature processing Peng et al. (2023), education (Peng et al., 2023), and clinical decision support (Thompson et al., 2023; Shi et al., 2023). Among these, clinical decision-making tasks place higher demands on the reasoning capabilities of models (Tordjman et al., 2025). RAG enhances LLMs through external knowledge, and the type of retrieval source significantly influences the quality of the generated outputs. Existing RAG approaches can be broadly categorized into two types based on the retrieval source (Gao et al., 2023): unstructured data and structured data. The former primarily consists of text and is widely used as a retrieval source, such as Wikipedia, PubMed, and similar resources. The latter includes tables or knowledge graphs, which typically provide more precise and structured information (Zhao et al., 2025). However, clinical decision-making tasks demand both supplementary information and strong reasoning. Traditional retrieval sources mainly provide information but offer little help for reasoning. Inspired by the inherent interpretability of

MDTs and their widespread use in CDSS, we propose an innovative approach that leverages MDTs as the retrieval source for RAG in medical reasoning scenarios.

**Medical Decision Trees.** MDTs are fundamental components of CDSS, representing explicit reasoning chains for clinical decisions (Podgorelec et al., 2002; López-Vallverdú et al., 2012). However, despite their advantages, leveraging MDTs directly as a structured retrieval source in RAG faces significant challenges: publicly available MDTs are scarce, and manually annotating the vast number required is impractical. To build MDTs automatically, Zhu et al. (2024a) proposed a Text-to-MDT dataset and introduced two extraction methods. He et al. (2024) proposed using generative models to extract MDTs and conducted experiments on the dataset published by Zhu et al., achieving promising results. However, both approaches have practical limitations due to the short length of the medical texts they used, where the decision-making process is highly concentrated. In fact, these texts themselves already represent clear decision-making processes that current LLMs can easily understand. In contrast, this paper focuses on extracting MDTs from long-form medical guidelines, where decision-making processes are distributed across sections such as pathological analysis, medication instructions, and contraindications. To tackle this challenge, we propose a novel method specifically designed for processing such complex, lengthy documents.

## 3 METHOD

In this section, we present our methodology, which focuses on two key aspects: constructing the retrieval source and enhancing LLMs with MDTs for medical reasoning.

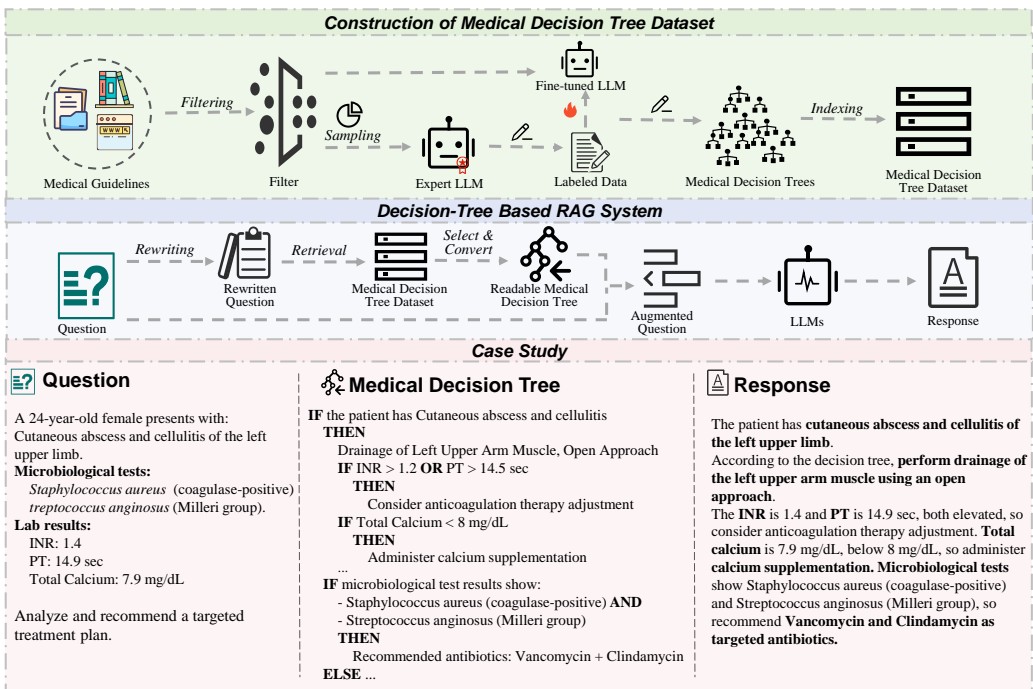

Figure 2: Overall construction of our MDT-RAG system. The top part shows the construction of the MDT dataset, where we use LLMs to extract large-scale MDTs for subsequent retrieval. The middle part illustrates the pipeline of MDT-RAG, which includes key steps such as query rewriting, retrieval, selection, and augmentation. The bottom part provides a case study demonstrating how MDTs guide medical reasoning.

## 3.1 Construction of Retrieval Source

This section introduces the construction of the retrieval source used in this experiment. We recognized the potential of MDTs in addressing medical reasoning problems; thus, we proposed enhancing medical LLMs with MDTs. Specifically, we extracted MDTs from medical texts such as PubMed, which form our retrieval source.

**Processing of Raw Information.** The primary texts we used were medical guidelines, which are evidence-based recommendations developed to assist healthcare professionals in making clinical decisions. Following Chen et al. (2023) and Wu et al. (2024), the data is sourced from a wide range of authoritative platforms, including StatPearls[2], NICE[3], PubMed[4], CDC[5], and others. Since our research focuses on medical reasoning scenarios, particularly those related to diagnosis and treatment, we filtered these texts based on their themes, retaining only those that met our requirements. The final dataset consists of 15,960 documents. Detailed information on data sources and statistics can be found in Figure 3.

**Data Annotation and Correction.**
Building such an RAG system requires a large number of MDTs as the retrieval source, and manual annotation would be prohibitively costly and time-consuming. Therefore, we adopted an automated approach for annotation. Given DeepSeek-R1's strong reasoning capabilities (Guo et al., 2025), we prompted it as an expert model to generate MDTs based on the medical guidelines. We observed that, while generative language models excel at producing natural language, they struggle

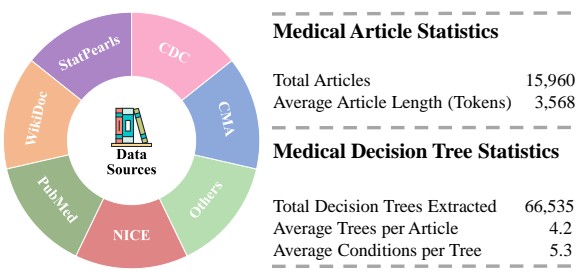

| Medical Article Statistics | |
| --- | --- |
| Total Articles | 15,960 |
| Average Article Length (Tokens) | 3,568 |

| Medical Decision Tree Statistics | |
| --- | --- |
| Total Decision Trees Extracted | 66,535 |
| Average Trees per Article | 4.2 |
| Average Conditions per Tree | 5.3 |

Figure 3: Medical article and decision tree statistics.

with generating recursively nested structures like MDTs. Particularly, as the recursive depth increases, the chance of errors in the generated trees grows significantly. To mitigate this issue, we opted for multi-branch MDTs instead of conventional binary trees, effectively reducing recursion depth. The correction of model-annotated trees is divided into two aspects: structural correctness and content correctness. The former ensures that the structure of the MDT strictly adheres to its definition. For the latter, we referred to the Fact Verification method used in knowledge graphs (Zhang et al., 2024; Quelle & Bovet, 2024), converting the MDT into natural language and leveraging the inherent knowledge of LLMs to verify and correct the content of the MDT. Finally, we obtained 2,400 annotated data entries.

**Supervised Fine-Tuning.** Due to the high cost of the expert model, we used the annotations generated by it to perform Supervised Fine-Tuning (SFT) on a smaller model $M_\theta$. The objective of this SFT stage is to train $M_\theta$ to accurately extract MDTs at scale. Specifically, we selected Qwen2.5-7B-Instruct-1M (Yang et al., 2025) as the fine-tuned model. Using the fine-tuned $M_\theta$, we processed about 16k clinical guidelines to extract MDTs. This process yielded a total of about 67k MDTs, which formed the core retrieval source of our MDT-RAG framework.

To evaluate the performance of $M_\theta$ in MDT extraction, we compared its results with those generated by an expert LLM on a test dataset. The findings show that $M_\theta$ achieves performance comparable to the expert LLM and even surpasses it on certain metrics. In terms of MDT structure accuracy, the expert LLM achieves 78.3%, while $M_\theta$ reaches 92.7%. This improvement is attributed to the corrected dataset used for fine-tuning, which ensures the structural correctness of the MDTs. For MDT content evaluation, we used BertScore (Zhang et al., 2019) to measure the similarity between the MDTs generated by $M_\theta$ and those produced by the expert LLM. The resulting score is 88.1%,

---

[2]www.statpearls.com

[3]www.nice.org.uk

[4]www.pubmed.ncbi.nlm.nih.gov

[5]www.cdc.gov

indicating that $M_\theta$ can produce content closely aligned with that of the expert LLM. More details are provided in Appendix A.2.

**MDT Structure Definition.** Here we provide a general formulation of our MDT. Let the MDT be denoted as $\mathcal{T}$, where each node $n \in N$ is defined as a quadruple $n = (B, T, \phi, C)$. Here, $B \in \{0, 1\}$ is the node type identifier, with $B = 0$ representing a non-leaf node (condition node) and $B = 1$ representing a leaf node (decision node). The set of triples $T = \{t_k\}_{k \in K}$ represents the atomic medical facts or conditions associated with the node, where each triple $t_k = (h_k, r_k, e_k)$ consists of a head entity $h_k \in \mathcal{H}$ (e.g., symptoms, or diagnostic tests), a relation $r_k \in \mathcal{R}$ (a predefined set of medical logical relations), and a tail entity $e_k \in \mathcal{E}$ (e.g., test results, or treatment options). The logical expression $\phi \in \Phi$ defines the selection condition for non-root nodes. It is composed of logical operators $\{AND, OR, NOT\}$ applied to indices $k \in K'$ from the parent node's triples. The list $C = [n_1, n_2, ..., n_m]$ contains the child nodes of this node. The semantics of the MDT is defined as follows: For a non-leaf node $n$, the selection of its child node $n_i \in C$ depends on evaluating its logical condition $\phi_{n_i}$. Specifically, $n_i$ is selected if and only if its logical expression $\phi_{n_i}$, which is defined over the triples $T_n$ of the parent node $n$, evaluates to true given the current context.

The tree is traversed as follows: Starting from the root node $\mathcal{R}$, at each non-leaf node encountered, evaluate the logical conditions $\phi$ of its child nodes based on the node's own triple set $T$. Then, move to the child node $n_i$ whose condition $\phi_{n_i}$ evaluates to true. This process repeats recursively until a leaf node is reached. At this point, all triples on the activated paths jointly form the final decision basis.

## 3.2 MDT-ENHANCED MEDICAL REASONING

We proposed a method that uses MDTs to improve the reasoning capabilities of LLMs in the medical domain. We extracted MDTs from clinical guidelines, and these MDTs formed the retrieval source. Compared to lengthy and scattered medical texts, MDTs provide a focused and clear analysis process. As shown in Figure 2, the process of enhancing LLMs with MDTs mainly includes three parts: MDT retrieval, filtering, and augmentation.

**MDT Retrieval.** We have extracted a large number of MDTs from clinical guidelines. Each MDT focuses on specific topics, such as disease diagnosis, medication usage, or treatment plans. For a given medical question, we need to identify the relevant MDTs and use them to enhance LLMs. Common retrieval approaches are designed for unstructured text or highly structured data. For unstructured text, retrieval methods can be divided into dense retrieval and sparse retrieval. Dense retrieval compares text vector similarity to capture complex semantic information (Devlin et al., 2019; Xiao et al., 2023), while sparse retrieval relies on keyword matching (Robertson et al., 2009). For structured resources like knowledge graphs, we can use query languages similar to SQL to perform precise retrieval, targeting specific nodes in the graph (Wang et al., 2023b; Pan et al., 2024). However, although MDTs are also structured, they are not suitable for retrieval using such query languages. On one hand, the structure of MDTs lacks a formal schema compared to knowledge graphs. On the other hand, MDTs represent the decision-making process, so extracting individual nodes is insufficient; a complete decision path is required for meaningful reference. Directly applying unstructured text retrieval methods is also problematic because the extracted MDTs contain substantial structural information, which can distort text similarity calculations and degrade retrieval performance. Therefore, we first converted the structured MDTs into natural language summaries containing key information, then transformed these summaries into embeddings and stored them in a vector database as indices for subsequent retrieval.

**MDT Filtering.** Previously, we introduced how to retrieve MDTs. After retrieving MDTs, we further filter them to ensure that the retrieved trees are useful for solving the problem. An MDT consists of a series of reasoning conditions and conclusions. We found that relying solely on dense retrieval can only preliminarily assess relevance. However, it cannot guarantee that the reasoning conditions and conclusions of the retrieved MDTs are applicable to the specific problem. Therefore, we need to further analyze whether an MDT's conditions match the patient's situation and whether its conclusions answer the core question. Because MDTs are structured data, traditional reranking methods and similarity calculations designed for plain text are not directly applicable. Thus, we

leverage the understanding capabilities of LLMs by prompting them to determine whether an MDT is helpful for answering the question, and based on this, we filter the retrieved MDTs.

**Using MDT to Enhance LLMs.** The retrieved MDTs are nested structures, making them unsuitable for directly augmenting LLMs, especially smaller models, which may struggle to understand such nested structures. Therefore, we need to convert these MDTs into natural language that is easier to comprehend. Earlier, we discussed the structure of MDTs. Converting such an MDT into natural language involves handling its triples and conditional judgments. For the triples, since our extraction focuses on disease diagnosis, medication usage, or treatment plans, the topics of the MDTs are relatively specific, allowing us to use predefined templates to translate the triples into natural language. For conditional judgments, consider a node $n$ and its child node $n_i$. The condition $\phi$ represents the requirement for moving from $n$ to $n_i$. In the MDTs, $\phi$ is an expression over indices of atomic conditions, combined with logical operators $\{AND, OR, NOT\}$ and parentheses. These expressions can sometimes be complex and not concise. Therefore, we first convert them into Disjunctive Normal Form (DNF), which resembles a "choose one of multiple options" structure, aligning better with natural language. Then, we represent the conditional judgments using IF-ELSE statements. After converting the MDTs into a natural representation, we incorporate them into the context to help LLMs answer medical questions.

## 4 EXPERIMENT

In this section, we describe the experimental setup and evaluation of our method. We first introduce the construction of our medical reasoning dataset, followed by the experimental design and comparative analysis with existing approaches.

### 4.1 DATASET

Although medical reasoning is a very common scenario, datasets in this field are extremely scarce. To the best of our knowledge, DDXPlus proposed by Fansi Tchango et al. (2022) is the only open-source medical reasoning dataset. However, this dataset is not suitable for evaluating generative language models. The dataset's core task requires diagnosing diseases based on symptoms and medical history. As a synthetic dataset, although it contains 1.3 million entries, it only covers 49 diseases. Consequently, its evaluation scope is narrow, failing to comprehensively assess the true capabilities of generative language models in broader medical reasoning tasks.

**Dataset Construction.** Therefore, we constructed a dataset focused on medical reasoning problems to test the reasoning abilities of LLMs. The dataset is derived from MIMIC-IV (Johnson et al., 2023), a real-world electronic health record dataset. Our dataset includes 2,046 entries, with each entry containing two medical questions. Specifically, we extracted the patient's clinical presentation, including symptoms, diagnoses, laboratory test results, and microbiology test results, which constitute the patient's background information. Then, we evaluated the clinical reasoning abilities of LLMs across two dimensions: medication recommendation and treatment plan design. The former requires LLMs to analyze indications, con-

Table 3: Comparison between DDXPlus and Our Dataset

| Property | DDXPlus | Ours |
|---|---|---|
| Disease Coverage (ICD) | 49 | 890 |
| Number of Instances | 1300k | 2046 |
| Avg. Question Length | 151 | 303 |
| Avg. Answer Length | 4.3 | 50.1 |
| Lab/Microbe Results | × | ✓ |
| Data Type | Synthetic | EHR |
| Primary Task | Diagnosis | Diagnosis & Treatment |

traindications, interactions, and individual patient differences, such as selecting appropriate medications based on the patient's medical test results while excluding contraindicated medications. This focuses on testing LLMs' ability to synthesize multi-source information and make precise decisions. The latter emphasizes overall planning, and since it generally involves multiple steps, it can be used to assess the step-by-step planning ability of medical models. Both question types are accompanied by reference answers extracted directly from the original medical records. Detailed statistics of our dataset can be found in Table 3.

Table 1: Performance for medication recommendations, focusing on multi-source synthesis and decision precision. The top two results in each row are highlighted in **bold** and underlined.

| Metric | Model | OQ | CoT | OQ-RAG | QR-RAG | Q2D-RAG | Tree-RAG |
|---|---|---|---|---|---|---|---|
| LQR(↓) | Llama-3.1-8B-Instruct | 96.0 | 26.0 | 23.0 | **17.0** | 18.0 | 18.0 |
| | Mistral-7B-Instruct-v0.3 | 97.5 | 84.5 | 75.0 | 75.0 | 75.0 | **66.0** |
| | Baichuan-M1-14B-Instruct | 79.5 | **3.0** | 16.5 | 7.5 | 9.5 | 7.0 |
| | Llama-3.3-70B-Instruct | 83.5 | **7.5** | 11.5 | 8.0 | 8.5 | 16.5 |
| | GPT-4o-mini | 75.0 | 3.0 | 3.5 | **2.5** | 3.5 | 7.5 |
| | Avg | 86.3 | 24.8 | 25.9 | **22.0** | 22.9 | 23.0 |
| AVG(↑) | Llama-3.1-8B-Instruct | 40.6 | 56.4 | 68.2 | 69.9 | 69.7 | **72.4** |
| | Mistral-7B-Instruct-v0.3 | 40.5 | 44.5 | 47.5 | 49.0 | 47.7 | **52.9** |
| | Baichuan-M1-14B-Instruct | 45.0 | 65.1 | 66.2 | 68.7 | 70.1 | **74.2** |
| | Llama-3.3-70B-Instruct | 43.6 | 71.7 | 76.8 | 77.1 | **77.9** | 76.8 |
| | GPT-4o-mini | 45.4 | 66.8 | 75.1 | 77.5 | 75.5 | **77.7** |
| | Avg | 43.0 | 60.9 | 66.8 | 68.2 | 68.2 | **70.8** |
| HQR(↑) | Llama-3.1-8B-Instruct | 0.0 | 2.5 | 9.0 | 11.0 | 13.0 | **15.0** |
| | Mistral-7B-Instruct-v0.3 | 0.0 | 3.0 | 1.5 | 2.5 | 2.0 | **5.0** |
| | Baichuan-M1-14B-Instruct | 0.0 | 2.5 | 8.5 | 5.5 | 5.0 | **11.0** |
| | Llama-3.3-70B-Instruct | 0.0 | 21.5 | 16.5 | 11.0 | 18.0 | **26.0** |
| | GPT-4o-mini | 0.0 | 9.0 | 16.0 | 15.0 | 11.0 | **22.5** |
| | Avg | 0.0 | 7.7 | 10.3 | 9.0 | 9.8 | **15.9** |

Table 2: Performance for treatment plan design, emphasizing step-by-step reasoning and long-term planning.

| Metric | Model | OQ | CoT | OQ-RAG | QR-RAG | Q2D-RAG | Tree-RAG |
|---|---|---|---|---|---|---|---|
| LQR(↓) | Llama-3.1-8B-Instruct | 87.5 | 17.5 | 23.5 | 11.5 | 14.5 | **9.5** |
| | Mistral-7B-Instruct-v0.3 | 92.5 | 71.0 | 69.0 | 52.0 | 57.5 | **49.5** |
| | Baichuan-M1-14B-Instruct | 43.0 | 1.5 | 9.0 | **1.0** | 8.5 | 4.0 |
| | Llama-3.3-70B-Instruct | 55.5 | **1.0** | 5.5 | **1.0** | 2.0 | 4.0 |
| | GPT-4o-mini | 38.5 | **0.5** | **0.5** | 1.0 | 1.5 | 1.0 |
| | Avg | 63.4 | 18.3 | 21.5 | **13.3** | 16.8 | 13.6 |
| AVG(↑) | Llama-3.1-8B-Instruct | 42.9 | 60.0 | 68.2 | 74.1 | 71.5 | **76.9** |
| | Mistral-7B-Instruct-v0.3 | 41.6 | 47.8 | 50.2 | 57.7 | 56.1 | **60.4** |
| | Baichuan-M1-14B-Instruct | 53.5 | 73.4 | 68.3 | 76.0 | 72.8 | **80.3** |
| | Llama-3.3-70B-Instruct | 50.7 | 80.4 | 81.8 | 84.3 | 83.1 | **85.3** |
| | GPT-4o-mini | 54.7 | 72.8 | 80.6 | 83.8 | 81.4 | **84.4** |
| | Avg | 48.7 | 66.9 | 69.8 | 75.2 | 73.0 | **77.5** |
| HQR(↑) | Llama-3.1-8B-Instruct | 0.0 | 5.0 | 9.5 | 14.5 | 11.0 | **17.0** |
| | Mistral-7B-Instruct-v0.3 | 0.0 | 3.5 | 2.0 | 6.0 | 4.5 | **9.5** |
| | Baichuan-M1-14B-Instruct | 0.0 | 13.0 | 6.0 | 10.5 | 13.0 | **17.5** |
| | Llama-3.3-70B-Instruct | 0.0 | 38.0 | 24.5 | 28.0 | 25.0 | **38.5** |
| | GPT-4o-mini | 0.0 | 14.0 | 20.5 | 26.5 | 19.5 | **29.5** |
| | Avg | 0.0 | 14.7 | 12.5 | 17.1 | 14.6 | **22.4** |

## 4.2 EXPERIMENTAL SETUP

We compared our method with general RAG approaches across multiple LLMs. Model selection was based on factors such as model size, open-source status, and medical specialization. Specifically, we chose five models for the experiments: Llama-3.1-8B-Instruct (Grattafiori et al., 2024) and Mistral-7B-Instruct-v0.3 (Jiang et al., 2023) as small general-purpose models, Baichuan-M1-14B-Instruct (Wang et al., 2025) as a medical-specialized model, Llama-3.3-70B-Instruct (Grattafiori et al., 2024) as a large open-source model, and GPT-4o-mini (Hurst et al., 2024) as a commercial closed-source model. The baseline methods comprise two categories: non-retrieval and text-based retrieval. Non-retrieval baselines include No Retrieval and Chain-of-Thought (CoT) (Wei et al., 2022). The former provides LLMs with only the original question, while the latter employs a detailed reasoning framework, guiding LLMs through a step-by-step reasoning process to complete reasoning and answer questions. Text-based retrieval methods include Original Question (OQ-

RAG) (Xiong et al., 2024), Query Rewriting (QR-RAG) (Gao et al., 2023; Ma et al., 2023), and Query2Doc (Q2D-RAG) (Wang et al., 2023a). Original Question uses the raw question for retrieval, Query Rewriting rewrites the question into a retrieval-friendly format before conducting retrieval, and Query2Doc is a query expansion technique where LLMs are prompted to generate a relevant document based on the original question, then both the original question and the generated document are used for retrieval. Additional details on the baseline methods can be found in the Appendix.

Following the practices of Gu et al. (2024) and Zheng et al. (2023), we employed an LLM-as-Judge approach to design a multi-tiered evaluation metric. Specifically, we use OpenAI o4-mini (OpenAI, 2025) as the evaluation model, and to save costs, we conducted experiments on 200 samples. Our metric scores the LLMs' responses on a scale from 1 to 5. For a medical question, knowing only the final answer is far from sufficient. Compared to other domains, medicine places significant emphasis on interpretability, focusing not only on the final answer but also on the logic and reasoning behind the LLMs' responses. Overall, the more detailed the LLMs' responses and the more comprehensive their reasoning processes, the higher the score. Detailed scoring criteria can be found in the Appendix.

**Main Results.**     To evaluate the effectiveness of MDT-RAG, we conducted comprehensive experiments across five typical LLMs with varying scales and medical specialization. The main results are presented in Table 2 and Table 2, which utilizes three key metrics: Low quality rate (LQR), representing the rate of responses scoring 2 or lower and indicating limited or ineffective clinical analysis; High quality rate (HQR), representing the rate of responses scoring a perfect 5 and indicating the highest quality and most comprehensive clinical reasoning; and Average (AVG), representing the normalized average score. Analyzing these tables, we can draw the following key conclusions:

**(1) MDT-RAG consistently outperforms text-based retrieval methods in both medication recommendation and treatment planning.** The former represents multi-source information integration capabilities, while the latter reflects long-term planning capabilities. Across both tasks, MDT-RAG achieves the highest AVG and HQR scores. This demonstrates that structured MDTs enable more precise clinical reasoning than verbose text fragments.

**(2) MDT-RAG achieves more improvements in AVG and HQR than in LQR.** This demonstrates that MDT-RAG effectively enhances LLMs' average capabilities and their performance in generating high-quality responses. In our evaluation system, LQR indicates severely limited or ineffective clinical analysis, suggesting that our method is more effective for enhancing advanced reasoning than for reducing fundamental errors. This is likely because the retrieval sources are incomplete, making it difficult to find relevant references for certain questions.

**(3) Different LLMs benefit differently from MDT-RAG.** Smaller LLMs exhibit the most significant improvements in AVG and LQR. This indicates that, compared to text-based RAG, the MDT structure provides clearer and more comprehensible guidance. For specialized medical LLMs like Baichuan-M1-14B-Instruct and large general-purpose LLMs, which possess extensive pre-trained medical knowledge, the CoT approach can significantly enhance their analytical capabilities. In this case, text-based RAG primarily focuses on providing knowledge and thus offers limited assistance. However, our method can still improve LLMs' performance in generating high-quality responses.

## 5 ABLATION STUDY

To evaluate the design of MDT-RAG, we analyze the impact of its core components and explore the effect of varying the number of MDTs.

### 5.1 COMPONENTS OF MDT-RAG

To examine the effects of various components in MDT-RAG, we conduct an ablation study as shown in Table 4. Specifically, we analyze the impact of removing the following key components: TreeIndex, which converts MDTs into a retrievable index by generating natural language summaries and embeds them into a vector space; TreeNLGen, which transforms the structured logic of MDTs into natural language expressions to ensure that complex decision logic can be understood by LLMs and used to enhance reasoning; and LLM-Filter, which leverages the capabilities of LLMs to filter and assess the relevance of MDTs,

retaining only those helpful for answering the given question. The term "w/o" indicates versions where specific components are removed. Our key observations are as follows:

- Removing any of the three key components (TreeIndex, TreeNLGen, or LLMFilter) results in decreased performance, demonstrating that each component contributes to MDT-RAG's overall effectiveness.

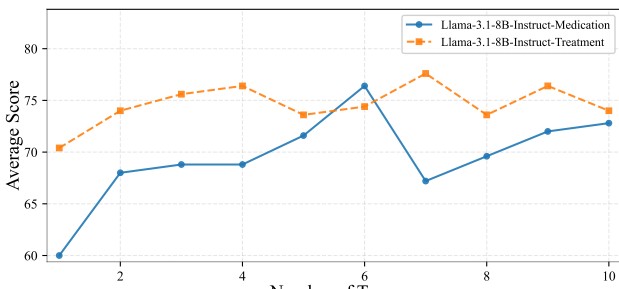

Figure 4: Impact of the Number of MDTs on LLM Performance (Average Score)

- A significant performance decline in MDT-RAG is observed when TreeIndex is removed. The original structure of MDTs is not suitable for retrieval; only by retrieving relevant MDTs can they effectively assist LLMs. LLMFilter also plays a significant role. As expected, the dense index alone is insufficient to identify relevant MDTs; we still need to leverage the capabilities of LLMs to filter. In addition, converting structured MDTs into clear and understandable natural language also proves to be helpful.

## 5.2 NUMBER OF MDTs

Next, we investigate the impact of the number of MDTs retrieved and used by MDT-RAG. We compare the effects of varying numbers of MDTs on the performance of MDT-RAG. In previous experiments, we set the number of MDTs to 6, which led to the token count of the input being reduced to approximately half that of text-based RAG. Through this exploration, we find that MDT-RAG performs best when the number of MDTs is moderate. When too few MDTs are used, they often fail to provide sufficient information. Conversely, as the number of MDTs increases, further improvement in LLMs' performance becomes difficult. This may be because the information in MDTs is highly concentrated, and an excessive number of them can interfere with LLMs' analysis and reasoning. Additionally, more MDTs increase the input token length, resulting in higher computational demands.

Table 4: Ablation study on different components of MDT-RAG, evaluated using Llama-3.1-8B-Instruct

| Method | Medication | | | Treatment | | |
|---|---|---|---|---|---|---|
| | LQR↓ | AVG↑ | HQR↑ | LQR↓ | AVG↑ | HQR↑ |
| Origin | 18.0 | 72.4 | 15.0 | 9.5 | 76.9 | 17.0 |
| w/o LLMFilter | 23.5 | 66.6 | 11.0 | 17.0 | 72.5 | 17.0 |
| w/o TreeIndex | 27.5 | 63.3 | 9.0 | 25.5 | 62.6 | 5.5 |
| w/o TreeNLGen | 23.5 | 68.4 | 11.5 | 14.0 | 73.0 | 12.0 |

## 6 CONCLUSION

In this paper, we propose MDT-RAG, the first RAG framework that uses MDTs for medical reasoning. Benefiting from the inherent interpretability of MDTs and their clear presentation of reasoning, MDT-RAG demonstrates superior performance over text-based RAG in medical reasoning tasks. To support the development of MDT-RAG, we introduce a high-quality MDT dataset, representing the largest open-source collection of its kind. To evaluate how effectively RAG systems enhance LLMs in medical reasoning, we developed a specialized benchmark dataset focused on information integration and long-term planning. Evaluation results demonstrate that MDT-RAG significantly enhances medical reasoning capabilities in LLMs. Our work demonstrates MDTs' potential as a powerful data structure for RAG systems, offering valuable insights to optimize RAG systems for complex reasoning tasks.

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

# A    RETRIEVAL DATASET CONSTRUCTION

In this section, we present the construction of the retrieval source and evaluation dataset. The retrieval source comprises numerous MDTs for our MDT-RAG system to query. The evaluation dataset is specifically designed for medical reasoning tasks to assess the RAG system's performance.

## A.1    RETRIEVAL SOURCE

This chapter describes the construction of retrieval sources. We collected medical guidelines from multiple sources and further filtered the articles to obtain moderately sized medical guidelines relevant to diseases. Clinical guidelines are rigorously researched frameworks designed to guide healthcare practitioners and patients in making evidence-based decisions regarding diagnosis, treatment, and management. They are compiled through a systematic process of collaborative consensus between experts to establish recommendations from the latest evidence on best practices that would maximize benefits in light of practical concerns such as available resources and context. These guidelines contain medical decision-making content. On one hand, they serve as the extraction source for MDTs; on the other hand, they also function as the retrieval sources for the text-based RAG system.

Our medical guidelines are sourced from:

- **StatPearls.** StatPearls is a comprehensive healthcare education platform that also serves as an open dataset. It offers a vast array of medical articles, educational resources, and point-of-care tools created by thousands of expert contributors. These resources cover a wide range of healthcare topics.

- **WikiDoc.** WikiDoc is a comprehensive open-source knowledge platform covering diverse topics. We utilize its medical section. Compared to clinical practice-oriented guidelines from other sources, WikiDoc's data leans toward encyclopedic content with an emphasis on explaining fundamental concepts.

- **PubMed.** PubMed is a comprehensive database of biomedical literature, maintained by the National Center for Biotechnology Information (NCBI). It serves as an essential resource for researchers, clinicians, and students, providing access to current and historical biomedical research.

- **NICE.** The National Institute for Health and Care Excellence (NICE) is a UK-based authority producing evidence-based clinical guidelines and health technology assessments. It provides standardized recommendations to improve care quality and efficiency across the NHS, serving as a key resource for healthcare professionals and policymakers.

- **CMA.** The Canadian Medical Association (CMA) is a national professional association representing physicians in Canada. It provides resources, advocacy, and support for physicians, promoting the highest standards of medical practice and patient care across the country.

- **CDC.** The Centers for Disease Control and Prevention (CDC) is a national public health agency in the United States. It focuses on disease prevention, health promotion, and emergency response, providing guidelines and resources to protect public health and improve healthcare practices.

- **WHO.** The World Health Organization (WHO) is a specialized agency of the United Nations responsible for international public health. It provides leadership on global health issues, sets standards, and offers guidance to countries to improve health systems and outcomes worldwide.

The construction of our dataset follows the settings of Chen et al. (2023) and Wu et al. (2024). The sources are relatively diverse, providing a rich retrieval base for our RAG system.

**Data Filtering.**    The data we obtained from the aforementioned sources are not all relevant to medical reasoning, particularly WikiDoc, where a significant portion of the text is of a popular science nature and thus difficult to directly apply to reasoning tasks. Specifically, our work focuses

on clinical medical reasoning scenarios. Therefore, we prioritize texts that can provide substantive assistance in reasoning.

Our data filtering process is based on thematic selection. We employ Qwen2-1.5B-Instruct (Qwen et al., 2025), a lightweight LLM, to perform this filtering. Additionally, we exclude excessively lengthy guidelines by implementing a length threshold to ensure retrieved guidelines remain concise. We set the threshold at 21k tokens; in practice, fewer than 1% of guidelines were filtered out due to length. Ultimately, from approximately 47k initial guidelines, we retained around 16k entries after filtering. The detailed filtering results are illustrated in Figure 5.

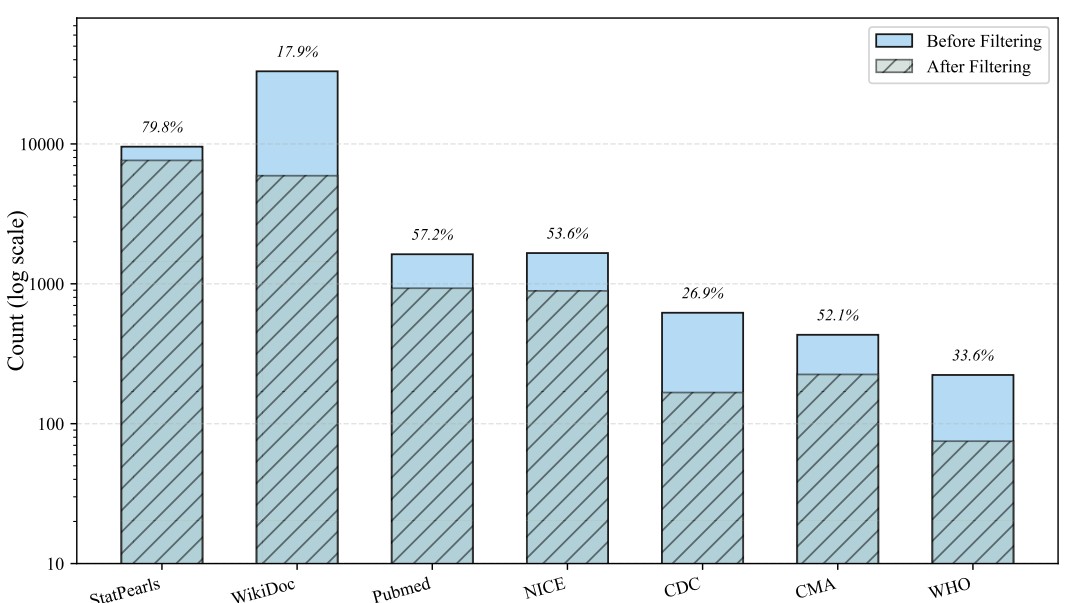

Figure 5: Data filtering results across different sources.

**Text-based RAG Source.** In the previous section, we described the collection and filtering of the retrieval dataset. In this chapter, we present the construction of the retrieval corpus for text-based RAG. We employ bge-base-en-v1.5 (Xiao et al., 2023) as the dense retrieval embedding model and partition the retrieval corpus into chunks with a *chunk_size* of 512 and an *overlap* of 100. For each retrieval operation, we retrieve 3 chunks and concatenate them into a single text segment to enhance the input for LLMs.

A.2  MDT-RAG SOURCE

In this section, we present the methodology for constructing the retrieval source in our MDT-RAG framework. We first extract MDTs from the pre-processed clinical guidelines, then create a searchable index of these MDTs to serve as the knowledge base for retrieval. During operation, we retrieve relevant MDTs and converts them into natural language representations for enhanced LLM integration.

**Expert Model Labeling.** To reduce annotation costs, we employ an automated annotation approach. First, we use an expert LLM to annotate the guidelines, then fine-tune a lightweight LLM with these annotations for subsequent labeling tasks. Specifically, we utilize DeepSeek-R1 (Guo et al., 2025), an LLM proficient in reasoning, as the expert LLM for annotation. Ultimately, we annotated approximately 2,400 samples, which still require further correction.

**Data Correction.** We observed that although DeepSeek-R1 exhibits strong reasoning capabilities, its annotations still contain certain errors. We categorize these errors into two types: factual

errors and structural errors. For the former, we adopt fact verification methods from the knowledge graph domain, leveraging LLMs to verify factual accuracy. For the latter, we implement automated checks to ensure that the MDT structure complies with our predefined specifications. If an MDT fails to meet the requirements, we re-annotate it using DeepSeek-R1 until all MDTs conform to the expected standards.

**Supervised Fine-tuning.** We use data annotated by DeepSeek-R1 to fine-tune a lightweight LLM for MDT extraction at scale. The LLM we fine-tuned is Qwen2.5-7B-Instruct-1M (Yang et al., 2025). The fine-tuning parameters are shown in Table 5. Thanks to the LoRA fine-tuning technique, our fine-tuning process can be completed on a single L20 GPU.

| Parameter | Max Length | Lora Alpha | Lora Dropout | Lora Rank | Target Module | Learning Rate | Epoch |
|---|---|---|---|---|---|---|---|
| **Value** | 20,000 | 32 | 0.05 | 32 | All | 5e-5 | 3 |

Table 5: Supervised Fine-tuning Parameters.

**SFT Labeling.** After fine-tuning, we employed the FT LLM $M_\theta$ to extract MDTs at scale. Ultimately, we extracted 66,535 MDTs from 15,960 guidelines. To evaluate the performance of $M_\theta$ in MDT extraction, we compared its results with those generated by an expert LLM. The findings indicate that $M_\theta$ achieves comparable or even superior performance to the expert LLM on certain metrics.

$M_\theta$ demonstrates excellent performance and produces high-quality MDTs. In terms of MDT structure accuracy, the expert LLM achieves 78.3%, while $M_\theta$ reaches 92.7%. This improvement stems from the corrected dataset used for fine-tuning, which ensures the structural correctness of the MDTs.

For MDT content evaluation, we measured multiple metrics including BertScore, triple F1, and reasoning path F1. Here, "triple" refers to atomic medical information units, while "reasoning path" denotes the complete sequence of conditions and conclusions from root to leaf nodes, composed of multiple triples.

The BertScore reaches 88.1%, initially suggesting that $M_\theta$ extracts MDT content comparable to the expert LLM. However, given that MDTs contain abundant structural information that may inflate BertScore, we additionally evaluated triple F1 and reasoning path F1. We employed the Sentence Transformer model gte-Qwen2-1.5B-instruct (Li et al., 2023) with a similarity threshold of 55 for evaluation. $M_\theta$ attained 81.8% triple F1 and 51.38% reasoning path F1, demonstrating its capability to effectively extract MDTs from medical texts.

To verify the correctness of the extracted decision trees, we referred to the Fact Verification method used in knowledge graphs (Zhang et al., 2024; Quelle & Bovet, 2024), converting the MDT into natural language and leveraging the inherent knowledge of medical LLMs to verify content of the MDT. Specifically, we randomly sampled 500 decision trees generated by the expert model and another 500 generated by $M_\theta$, and then evaluated them using medgemma-27b-text-it (Sellergren et al., 2025), a medical domain LLM. respectively. The results show that 95.8% of the decision trees extracted by the expert model and 93.6% of those extracted by the $M_\theta$ were recognized by the medical domain LLM.

**MDT Indexing.** To construct the retrieval source for MDT-RAG, after extracting the MDTs, we need to convert them into embedding vectors for retrieval. However, directly transforming the extracted MDTs into embeddings leads to poor retrieval performance because they contain substantial structural information, which interferes with retrieval. Therefore, we converted them into natural language descriptions for retrieval purposes.

In summary, we employ LLMs to summarize the MDTs, focusing primarily on disease symptoms, diagnosis, and treatment to facilitate subsequent retrieval and reasoning tasks. These summaries of MDTs are then used as embedding vectors for retrieval. Compared to the original guidelines, MDTs represent more condensed knowledge, eliminating the need for chunking—they can be retrieved as a whole. We utilize Qwen2.5-14B-Instruct-1M (Qwen et al., 2025) to generate the summaries of MDTs.

## B    EVALUATION DATASET CONSTRUCTION

In this section, we detail the construction of the evaluation dataset. With the advancement of LLMs, their reasoning capabilities have significantly improved. However, there remains a scarcity of existing medical reasoning datasets. Before LLMs emerged, reasoning was a highly challenging task. It was difficult to solve by using traditional machine learning methods. Early medical reasoning tasks tended to focus more on multi-class classification problems, such as ddx-plus, where the questions were essentially framed as classification tasks. However, today, LLMs are capable of handling complex reasoning tasks and are no longer satisfied with merely generating a final answer; instead, they require the reasoning process itself. Against this backdrop, we propose an evaluation dataset specifically designed for medical reasoning tasks.

### B.1    EVALUATION DATASET SOURCE

Our data is sourced from MIMIC-IV (Johnson et al., 2023), a widely used medical dataset containing extensive clinical records and healthcare information. MIMIC-IV is a publicly available medical database comprising patient data from a large hospital in the Boston area, USA. The dataset covers diverse diseases and treatment plans, including diagnostic information, treatment records, laboratory test results, and more. With its large volume and high diversity, MIMIC-IV provides a robust foundation for medical reasoning tasks.

We extracted approximately 41k records from MIMIC-IV, including patient age, gender, diagnostic information, medical test results, medications, procedures, and other relevant details. Further filtering was subsequently applied to refine the dataset.

### B.2    EVALUATION DATASET FILTERING

After extracting data from MIMIC-IV, we further filtered the dataset. Since our goal was to construct an evaluation dataset specifically for medical reasoning tasks, we needed to ensure that the questions in the dataset had sufficient complexity and diversity. Therefore, during the filtering process, we prioritized data entries that included more comprehensive medical test results and well-documented quality protocols. Ultimately, we curated an evaluation dataset containing approximately 2k records.

For each patient's information, we designed two types of questions: medication recommendation and treatment plan recommendation. The former focuses on the ability to integrate multi-source information, while the latter assesses long-term planning capabilities. Our filtering process ensured that both types of questions had reference answers available.

## C    IMPLEMENTATION DETAILS

In this section, we present the implementation details of the MDT-RAG system, including the retrieval process and how the retrieved MDTs are utilized to enhance LLMs.

### C.1    MDT RETRIEVAL

In this section, we describe the retrieval process of the MDT-RAG system. Drawing on the query rewriting technique from text-based RAG, we summarize medical questions into key symptoms and medical test indicators to optimize the query. Notably, our query rewriting here is the same as that of the baseline text-based RAG method. After obtaining these queries, we convert them into embedding vectors and then use vector retrieval to search for MDTs.

### C.2    TREENLGEN

In the previous section, we introduced the retrieval process of the MDT-RAG system. However, the retrieved MDTs remain structured MDTs, which are not LLM-friendly. Specifically, these structured MDTs contain extensive structural information that may hinder LLMs' comprehension. Moreover, the original MDTs represent node relationships through nested parentheses, a format that is also unfriendly to LLMs. Therefore, we need to convert these MDTs into natural language descriptions for retrieval.

To convert structured MDTs into natural language, an intuitive approach is to employ LLMs for processing. However, we found that lightweight LLMs perform suboptimally when handling such structured MDTs. While larger-scale LLMs could be utilized, this would significantly increase system overhead. Consequently, we opted to use predefined templates to transform structured MDTs into natural language descriptions. Specifically, we designed a template primarily responsible for converting MDT triples into natural language. Given our focus on medical reasoning, the scope of our MDTs is also confined to relevant domains. The triples in these MDTs mainly encompass symptoms, medical tests, diagnoses, and treatments. This limited thematic range enables us to design templates for effectively translating MDT triples into natural language.

### C.3 MDT Filtering

The performance of RAG systems heavily relies on the quality of retrieved text. Therefore, after retrieving MDTs, we conducted further filtering. Here, we leveraged the comprehension capabilities of LLMs to determine whether retrieved MDTs were helpful for answering medical reasoning questions. Specifically, we obtained 25 MDTs during the retrieval phase and then used LLMs for filtering. We employed two filtering approaches:

First, we directly used an LLM to analyze retrieved MDTs, where these 25 trees were directly input into LLMs. We utilized Qwen2.5-14B-Instruct-1M (Yang et al., 2025) for this filtering.

The second approach involved prompting gte-Qwen2-7B-instruct (Li et al., 2023) to evaluate the helpfulness of each MDT for the question. We then set a threshold and selected trees scoring above this threshold as the final retrieval results. Both methods achieve similar effects.

## D BASELINE SETTINGS

In this section, we present the baseline setup, including specific implementation details. Our comparative approaches include non-retrieval methods and text-based retrieval methods, with implementations as follows.

### D.1 NON-RETRIEVAL BASELINE

In this section, we present the non-retrieval baseline, where LLMs directly answer questions without employing any retrieval methods. This approach primarily serves to evaluate the intrinsic capabilities of LLMs, independent of external knowledge.

- **No Retrieval.** This baseline employs LLMs to directly answer questions without utilizing any retrieval methods. It also does not employ prompt engineering techniques, providing only the raw question to the LLM without any guiding prompts.
- **Chain-of-Thought (Wei et al., 2022).** Chain-of-Thought (CoT) refers to a structured reasoning approach that decomposes complex problems into sequential logical steps. By explicitly modeling the step-by-step thought process, it enhances both transparency and accuracy in problem-solving. In our setup, we prompt the LLM to follow standard medical analysis procedures: first analyzing patient symptoms and medical test results individually, then determining the patient's condition based on these diagnostic findings, and finally proposing a treatment plan.

### D.2 TEXT-BASED RAG BASELINE

In this section, we present the baseline of text-based RAG. This baseline employs a text retrieval approach to answer questions by retrieving relevant medical literature. The baselines we compare include the original text-based RAG method, query rewriting, and query expansion. The specific configurations are as follows.

- **Original Question (Xiong et al., 2024).** This baseline uses the original question for retrieval without any modifications, serving to analyze the retrieval effectiveness of the original question.

- **Query Rewriting (Gao et al., 2023; Ma et al., 2023).**   Query Rewriting reformats the question into a retrieval-friendly version before conducting retrieval. In our setup, we employ LLMs to summarize medical reasoning questions into key symptoms and medical test indicators, optimizing the query. The configuration used here is identical to the query rewriting in our MDT-RAG framework.

- **Query2Doc (Wang et al., 2023a).** Query2Doc is a query expansion technique where the LLM generates a relevant document based on the original question, and both the original question and the generated document are used for retrieval. This method enhances retrieval comprehensiveness by expanding the query with generated documents. For consistency, we uniformly use Qwen2.5-7B-Instruct to generate the documents here.

Notably, the prompts used in text-based RAG and MDT-RAG are identical, with the only difference being that the former fills the prompt with retrieved text fragments while the latter uses retrieved MDTs. This design ensures a fair comparison between the two approaches.

# E  COST ANALYSIS

In this section, we analyze the cost of MDT-RAG. The cost of constructing an MDT-RAG system primarily consists of two aspects: one is the extraction cost of MDTs during retrieval construction, and the other is the cost during inference with MDTs.

## E.1  MDT EXTRACTION COST

During the extraction of MDTs, we employed an expert LLM, which incurs a high extraction cost. However, after the MDTs are extracted, we fine-tuned a lightweight LLM, which has a significantly lower fine-tuning cost. Therefore, the MDT extraction cost for the MDT-RAG system includes the annotation cost of the expert LLM and the fine-tuning and annotation costs of the lightweight LLM.

We used the DeepSeek-R1 for MDT extraction, annotating a total of 2400 data points for fine-tuning and testing. Statistics show that the input was approximately 11.1M tokens and the output was approximately 8.7M tokens. Based on the DeepSeek-R1 pricing of $0.57 per million tokens for input and $2.3 per million tokens for output, the total cost for MDT extraction was $17.3 USD.

For fine-tuning the lightweight LLM, we used the Qwen2.5-7B-Instruct-1M. Fine-tuning was performed on a single L20 GPU and took approximately 5 hours. During the annotation phase, we also used an L20 GPU for annotation, which took approximately 6 hours.

## E.2  MDT-RAG COST

In this section, we analyze the cost of MDT-RAG. In our setup, MDT-RAG employs up to six MDTs to assist reasoning, while the text-based RAG method uses three text fragments of 512 tokens each for reasoning. The MDT-augmented queries average 1.1k tokens, whereas the text-based RAG augmented queries average 1.9k tokens. Overall, MDT-RAG incurs some additional overhead during the construction phase. However, once deployment is complete, the required computational resources will be significantly reduced.

# F  CASE STUDY

In this section, we introduce several MDT cases extracted from medical guidelines and demonstrate their application within the MDT-RAG system. As mentioned in the main text, multiple MDTs can be extracted from a single medical guideline, and these MDTs vary in size. Due to space constraints, we present several medium-sized MDTs here, with only one MDT showcased per guideline.

# Case 1

## *Medical guideline*

Sphincter of Oddi Dysfunction

...
Introduction
... In a diagnosis of exclusion, the typical patient presents with recurrent biliary colic-type symptoms, generally after undergoing cholecystectomy, often in concert with transaminitis, pancreatitis, or both...
Evaluation
...
Specific diagnostic criteria for SOD include:
Transaminitis (greater than 2 times the upper limit of normal on 2 or more occasions) Common bile duct dilation (greater than 10 mm on US; greater than 12 mm on ERCP) Biliary pain
Utilizing these criteria, patients are classified as follows:
Type I SOD: all 3
Type II SOD: biliary pain and 1 of the other 2 criteria.
Type III SOD: biliary pain only [3]
The results of this classification impact the subsequent treatment plan.
...
Treatment / Management
...
Noninvasive options include calcium channel blockers, tricyclic antidepressants, glyceryl trinitrate, and somatostatin...
Invasive interventions for the treatment of SOD include ERCP with sphincterotomy. ... Patients with type I and type II SOD should be referred for management with ERCP and sphincterotomy. Type III SOD has been shown in a trial not to respond to procedural intervention. [9] These patients should instead be referred for medical management, including pain control, as discussed above.

## *MDT in Natural Language*

**IF**: ( Patient has Clinical manifestations of recurrent biliary colic-type symptoms ) **AND** ( Patient has Past treatment history of cholecystectomy ) **AND** ( Diagnosis is of exclusion of other pathologies ) :

   **IF**: ( Patient has Clinical manifestations of biliary pain ) **AND** ( Diagnostic test results show transaminitis )
      **AND** ( Diagnostic test results show CBD dilation ) :
     Treatment plan includes ERCP with sphincterotomy

   **IF**: ( Patient has Clinical manifestations of biliary pain ) **AND** ( Diagnostic test results show transaminitis    **OR** Diagnostic test results show CBD dilation ) :
     Treatment plan includes ERCP with sphincterotomy

   **IF**: ( Patient has Clinical manifestations of biliary pain ) **AND** ( Diagnostic test results show no signs of transaminitis ) **AND** ( Diagnostic test results show no signs of CBD dilation ) :
    calcium channel blockers is prescribed for Treatment medications
    tricyclic antidepressants is prescribed for Treatment medications
    glyceryl trinitrate is prescribed for Treatment medications
    Treatment plan includes combination therapy

## *MDT*

Figure 6: MDT Case Study 1

## Case 2

### *Medical guideline*

Amebic Liver Abscess

Introduction
Amebiasis is a parasitic infection caused by the protozoan, Entamoeba histolytica; transmitted through fecal-oral route... Optimal treatment includes the use of Metronidazole followed by a luminal agent such as Paromomycin.
Etiology
Entamoeba histolytica is a parasitic protozoan that is a common cause of amoebic colitis...
...
Treatment / Management
Treatment entails the use of a Nitroimidazole, preferably Metronidazole at a dose of 500 mg to 750 mg by mouth 3 times per day for 7 to10 days. Alternatively, Tinidazole 2 gm by mouth daily for 3 days can be used. Since the parasites can persist in the intestine in 40% to 60% of patients, treatment with a Nitroimidazole should always be followed with a luminal agent such as Paromomycin 500 mg 3 times a day for 7 days... Around 15% of patients with amebic liver abscess fail medical treatment. Therapeutic aspiration can be done either by percutaneous needle aspiration or by percutaneous catheter drainage. These options should be considered in patients with no clinical response to antibiotics within 5 to 7 days, in those with a high risk of abscess rupture (cavitary diameter over 5 cm or presence of lesions in the left lobe), or in cases of bacterial coinfection of amebic liver abscess. [10] Between percutaneous needle aspiration and percutaneous catheter drainage, studies have shown that the latter is superior with higher success rate and quicker resolution.
...

### *MDT in Natural Language*

**IF**: ( Abscess size >5 cm **OR** Abscess location left lobe **OR** Patient has bacterial coinfection ) :
    Treatment plan includes percutaneous catheter drainage
    Paromomycin is prescribed for Treatment medications
**IF**: ( Abscess size not >5 cm ) **AND** ( Abscess location not left lobe ) **AND** ( Patient does not have bacterial
        coinfection ) :
    **IF**: ( Metronidazole is prescribed for Treatment medications **OR** Tinidazole is prescribed for Treatment
        medications ) :
    **IF**: ( Metronidazole is not prescribed for Treatment medications ) :
      Paromomycin is prescribed for Treatment medications
      **IF**: ( Treatment response to Nitroimidazole ) :
        Treatment plan includes percutaneous catheter drainage
        Paromomycin is prescribed for Treatment medications

### *MDT*

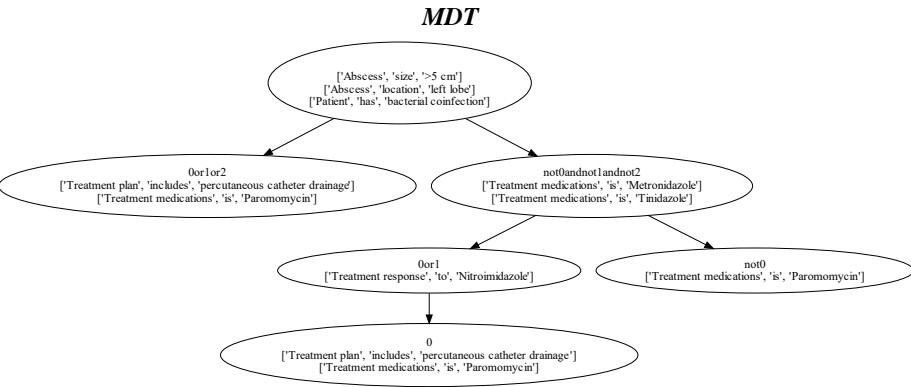

Figure 7: MDT Case Study 2

# Case 3

## *Medical guideline*

Nummular Dermatitis

Introduction
Nummular dermatitis, also called discoid eczema or nummular eczema, is a pruritic eczematous dermatosis characterized by multiple coin-shaped lesions. ... The prognosis of this condition is excellent. Most cases can be treated successfully with conservative measures and topical corticosteroids, and most patients will eventually achieve remission. ...

Complications
Because of the impaired skin barrier, lesional skin may become secondarily infected; staphylococcus aureus is the most commonly implicated pathogen. Impetiginized lesions can display purulent ooze and thicker golden crusting than noninfected lesions. A bacterial swab should be performed for culture and sensitivities. Based on local antimicrobial resistance patterns, doxycycline or another antistaphylococcal antibiotic may be selected initially; further treatment can be tailored according to the resultant sensitivities. ...

General Measures
Frequent moisturization with thick emollients such as petroleum jelly is recommended. ... Patients should be instructed to take short (≤5 minutes) lukewarm showers, use gentle hydrating soaps, and apply emollients immediately after showering while the skin is still slightly wet. ...

Topical Therapies
High- or ultrahigh potency topical corticosteroids applied directly to affected skin 1 to 2 times daily help decrease inflammation and pruritus. ...

Phototherapy
For widespread disease in which topical treatment may not be feasible, narrowband UVB light therapy should be considered. ...

Systemic Therapies
... If signs of a secondary bacterial infection or a bacterial swab of lesional skin are positive, treatment with topical or oral antistaphylococcal antibiotics is recommended, depending on whether lesions are localized or diffuse. ...

## *MDT in Natural Language*

**IF**: ( Lesion has Clinical manifestations of purulent ooze or golden crusting **OR** Lesion has Diagnostic test results of positive bacterial culture ) :
    Patient has Treatment medications of topical/oral antistaphylococcal antibiotics
**IF**: ( Lesion does not have Clinical manifestations of purulent ooze or golden crusting ) **AND** ( Lesion does not have Diagnostic test results of positive bacterial culture ) :
    **IF**: ( Lesion has Distribution of localized ) :
        Treatment plan includes topical corticosteroids
        Treatment plan includes emollients
    **IF**: ( Lesion has Distribution of widespread ) :
        Treatment plan includes narrowband UVB phototherapy

## *MDT*

Figure 8: MDT Case Study 3

**Prompt 1: Prompt of Decision Tree Extraction**

As an intuitive hierarchical decision-making tool, the decision tree makes the next condition judgment or decision based on the outcomes of different condition judgments. It can be used for disease diagnosis, treatment selection, and drug analysis. Now we introduce the multi-way medical decision tree, which starts at the root node and branches step by step through conditional judgments. Non-leaf nodes are condition nodes, while leaf nodes are decision nodes. Non-leaf nodes contain multiple conditions that can be judged, namely triple relationships. Based on these relationships, different child nodes can be selected, thereby generating new branches. For a leaf node, all its triple relationships must be satisfied. In this case, the triple relationships actually represent the final decision. All nodes contain a "select_condition" item, which indicates which conditions of the parent node are satisfied. In a medical decision tree, the parent node has triple relationships that can be judged. Satisfying different triple relationships leads to different decisions and, consequently, different child nodes. For non-leaf nodes, their "children" represent a list of child nodes. The structure of all nodes is the same. For leaf nodes, this field is an empty list.

The conditions that can be judged within non-leaf nodes are represented in the form of triple tuples, namely "head entity," "relationship," and "tail entity." The head entity is often a type of patient, a disease, a diagnostic test, a drug, etc. Here, we have defined some triple relationships for your reference. The relationships are listed below.

# Start of Relationship List

[ Clinical manifestations, # patient's symptom conditions, location of symptoms, severity of symptoms, etc.
Treatment medications, # drugs that can be used to treat the disease or patient
Dosage and administration, # dosage of treatment medications
Treatment plan, # treatment plan for the disease or patient
Contraindicated medications,
Basic information, # patient's age, gender, weight, etc.
Recommended department, # recommended hospital department for the disease or patient
Diagnostic tests, # recommended diagnostic tests for the disease or patient
Family history,
Travel history,
Medication history,
Past treatment history,
Imaging characteristics,
Diagnostic test results, # results of a specific medical diagnostic test
Duration of symptoms,
Complications, # any concurrent symptoms in addition to the main symptoms
Changes in appetite,
Changes in sleep quality,
Changes in weight,
Allergic status,
Substance use,# use of tobacco, alcohol, etc.
Treatment response # patient's response to the current treatment plan
] # End of Relationship List In addition to these relationships, you can also choose other relationships as needed. Each node in a decision tree can be represented as: { "is_leaf": "0" "1", # "0" indicates that the node is not a leaf node but a condition node, while "1" indicates that it is a leaf node.
"triples": {
"0": [head entity0, relation0, tail entity0],
"1": [head entity1, relation1, tail entity1],
...
}, # The extracted triple relationships. Note that these relationships must be conditions that can be judged as true or false, so that the corresponding child nodes can be selected based on which conditions are met. For leaf nodes, all triple relationships should be true, thereby preventing the creation of new branches.
"select_condition": ' ', # Indicates which triple relationships of the parent node are satisfied by this node. Use logical condition expressions with logical words "and", "or", "not", and parentheses. For example, if this node satisfies the parent node's conditions "0" and "3", it should be "0and3". If it satisfies "1" and "2" or only "3", it should be "(1and2)or3". If selecting this node requires the parent node's condition "1" to be true while condition "2" is false, it can be expressed as "1andnot2". Only when explicitly requiring a condition to be false can the "not" logic be used. For a root node, this field is an empty string; for non-root nodes, this field must not be empty.
"children": [child_node1,child_node2...], # The decision tree is represented through nesting. In the non-leaf nodes of the decision tree, the 'children' field represents its child nodes, and all nodes have the same structure. For leaf nodes, the 'children' field is an empty list, indicating that they have no child nodes.
}
Overall, our decision tree can be represented through the nesting of the nodes described above.

Your task is to extract decision trees from the specified medical text. The decision trees can revolve around themes such as disease diagnosis and treatment, drug analysis, etc. A medical text may contain more than one decision tree. If there are multiple decision trees, please reply with a dictionary of decision trees. The format should be {"tree1": the first decision tree you extracted, "tree2": the second decision tree you extracted, ...}. Please extract as many decision trees as possible. Do not reply with any content other than the decision trees. No annotations are needed either.
The text you need to process is:

---

**Prompt 2: Prompt of MDT-RAG**

As an AI researcher, I need to evaluate your medical capabilities. You will be provided with a question and some background information. The background information is sourced from medical literature and is considered accurate and reliable.
[Start of Background Information]
{context_str}
[End of Background Information]
The question you need to answer is:
[Start of Question]
{query_str}
[End of Question]
First, you need to analyze the patient's symptoms and medical test information one by one, explaining their meanings. Then, through differential diagnosis, determine the patient's condition. Next, provide the answer to the question and the rationale considering the background information. Finally, analyze more possible scenarios according to the background information.
Note that when using background information, repeat it and cite the source, e.g., 'article [insert article name] indicates that...' since evaluators cannot see the background information directly.

---

**Prompt 3: Prompt of Text-based RAG**

As an AI researcher, I need to evaluate your medical capabilities.
You will be provided with a question and some background information. The background information is sourced from medical literature and is considered accurate and reliable.
[Start of Background Information]
{context_str}
[End of Background Information]
The question you need to answer is:
[Start of Question]
{query_str}
[End of Question]
First, you need to analyze the patient's symptoms and medical test information one by one, explaining their meanings. Then, through differential diagnosis, determine the patient's condition. Next, provide the answer to the question and the rationale considering the background information. Finally, analyze more possible scenarios according to the background information.
Note that when using background information, repeat it and cite the source, e.g., 'article [insert article name] indicates that...' since evaluators cannot see the background information directly.

---

**Prompt 4: Prompt of CoT**

As an AI researcher, I need to evaluate your medical capabilities. You will be provided with a medical question.
The medical question you need to answer is:
[Start of Question]
{query_str}
[End of Question]
First, you need to analyze the patient's symptoms and medical test information one by one, explaining their meanings. Then, through differential diagnosis, determine the patient's condition. Next, provide the answer to the question and the rationale considering the background information. Finally, analyze more possible scenarios according to the background information.

1296
1297
1298
1299
1300
1301
1302
1303
1304
1305
1306
1307
1308
1309
1310
1311
1312
1313
1314
1315
1316
1317
1318
1319
1320
1321
1322
1323
1324
1325
1326
1327

### Prompt 5: Prompt of Tree Summary

You will be given a medical decision tree below. Analyze its content. If it mentions any diseases or symptoms, list them. If none are mentioned, reply only with 'None'. You only need to provide the name of the disease or symptom, or None, without any explanation or additional information. Please reply in English.
First, I will give you a few examples.
[First example begins]
IF: ( Patient has Clinical manifestations of localized pain upon palpation of medial epicondyle ) AND ( Patient has Clinical manifestations of history of repetitive handwrist flexion ) : Golfer's elbow has Treatment plan of nonsteroidal anti-inflammatories, Golfer's elbow has Treatment plan of rest,
[First example ends]
The answer for the first example is:
Localized pain upon palpation of medial epicondyle.
Repetitive hand/wrist flexion
Golfer's elbow
[Second example begins]
IF: ( Patient has Basic information of pediatric age ) :
IF: ( Patient has Basic information of weight 5-10 kg ) :
Omeprazole has Dosage and administration of 5 mg daily
IF: ( Patient has Basic information of weight 10-20 kg ) :
Omeprazole has Dosage and administration of 10 mg daily
IF: ( Patient has Basic information of weight ¿20 kg ) :
Omeprazole has Dosage and administration of 20 mg daily.
[Second example ends]
The answer for the second example is:
None
The decision tree you need to process is:
[Decision tree begins]
{tree}
[Decision tree ends]
Your reply:

1328
1329
1330
1331
1332
1333
1334
1335
1336
1337
1338
1339
1340
1341
1342
1343
1344
1345
1346
1347
1348
1349

### Prompt 6: Prompt of Tree verification

You will be given a piece of medical knowledge. Please determine whether it is correct. If it is correct, reply with T. If it is incorrect, reply with F. Then, please provide the reasoning.
[Medical Dicision Rule Start]
{rule}
[Medical Dicision Rule End]
Your answer:

Prompt 7: Template of TreeNLGen

{ "Clinical manifestations": [ "{head} exhibits {tail}", "{head} does not exhibit {tail}" ],
"Treatment medications": [ "{tail} is prescribed for {head}", "{tail} is not prescribed for {head}" ],
"Dosage and administration": [ "The dosage regimen for {head} is {tail}", "There is no established dosage regimen of {tail} for {head}" ],
"Treatment plan": [ "{head}'s treatment plan includes {tail}", "{head}'s treatment plan does not include {tail}" ],
"Contraindicated medications": [ "{head} should avoid {tail}", "{head} can safely take {tail}" ],
"Basic information": [ "{head}'s {relation} is {tail}", "{head}'s {relation} is not {tail}" ],
"Recommended department": [ "{head} should be referred to the {tail} department", "{head} does not require referral to the {tail} department" ],
"Diagnostic tests": [ "{head} requires {tail} testing", "{head} does not require {tail} testing" ],
"Family history": [ "{head} has a family history of {tail}", "{head} has no family history of {tail}" ],
"Travel history": [ "{head} has recent travel to {tail}", "{head} has no recent travel to {tail}" ],
"Medication history": [ "{head} has previously taken {tail}", "{head} has never taken {tail}" ],
"Past treatment history": [ "{head} received prior treatment with {tail}", "{head} has never received treatment with {tail}" ],
"Imaging characteristics": [ "Imaging of {head} shows {tail}", "Imaging of {head} shows no evidence of {tail}" ],
"Diagnostic test results": [ "{head} test results show {tail}", "{head} test results show no signs of {tail}" ],
"Duration of symptoms": [ "{head} has experienced symptoms for {tail}", "{head} has not experienced prolonged symptoms ({tail})" ],
"Complications": [ "{head} developed complications including {tail}", "{head} has not developed {tail} complications" ],
"Changes in appetite": [ "{head} reports changes in appetite: {tail}", "{head} reports no changes in appetite" ],
"Changes in sleep quality": [ "{head} reports changes in sleep pattern: {tail}", "{head} reports no changes in sleep pattern" ],
"Changes in weight": [ "{head} has experienced weight changes: {tail}", "{head} has not experienced significant weight changes" ],
"Allergic status": [ "{head} has allergies to {tail}", "{head} has no known allergies to {tail}" ],
"Substance use": [ "{head} uses {tail}", "{head} does not use {tail}" ],
"Treatment response": [ "{head} responds to treatment with {tail}", "{head} shows no response to treatment with {tail}" ],
"default": [ "{head} has {relation} of {tail}", "{head} does not have {relation} of {tail}" ]
}

---

**Prompt 8: Prompt of Scoring**

Below is a medical question, a reference answer, and an answer generated by a model. Please evaluate and score the model's response based on the given evaluation criteria. The reference answer provides only the final result without any reasoning, making it of limited reference value. In contrast, we prioritize detailed reasoning and explanations.
Medical Question:
[Medical Question Start]
{question}
[Medical Question End]
Reference Answer:
[Reference Answer Start]
{standard_answer}
[Reference Answer End]
Answer from Model:
[Model-Generated Answer Start]
{generated_answer}
[Model-Generated Answer End]
The scoring criteria are as follows:
1 point: Basic statement. The response did not analyze the patient's condition.
2 points: Elementary reasoning. It offers a simple explanation and conducts a basic analysis of the patient's specific situation.
3 points: Structured reasoning. It thoroughly analyzes the patient's symptoms and medical test results, performs preliminary differential diagnostics to rule out similar symptoms or manifestations, determines the patient's condition.
4 points: Systematic argumentation. The reasoning is relatively complete, taking into account the patient's symptoms, diagnostic information, and medical test results. Besides, medical literature is used to support the analysis, making the reasoning more credible and clinically valuable.
5 points: Advanced Reasoning. The reasoning is not limited to the information provided about the patient, but considers more possibilities, offering reasoning under a broader range of scenarios.
Since the patient's information may be incomplete, we place more emphasis on the diversity of analysis, and whether a wider range of scenarios have been considered.
Please score strictly according to the given scoring criteria.
Output only the scores in this exact format:
{"score":X}, where X is an integer between 1 and 5. Do not add any other text or explanations.

---

**Prompt 9: Prompt of Adding Title**

Below, you will be given a statement and a medical guideline. First, you need to determine whether the statement is the first sentence of the guideline or the title of the guideline. If it is the title, reply with the statement itself (i.e., the original title). If the statement is not the title, generate a title for the guideline (no more than 12 words) and reply with that title.
Only reply with the original title or the generated title—do not include any additional content.
Statement:
[Start of statement]
{statement}
[End of statement]
Guideline:
[Start of guideline]
{guideline}
[End of guideline]
Please provide your answer.
The title of the guideline is:

---

**Prompt 10: Prompt of Query2Doc**

Write a passage that answers the given query:
Here are a few examples for your reference.
[Examples Start]
Query: what state is this zip code 85282
Passage: Welcome to TEMPE, AZ 85282. 85282 is a rural zip code in Tempe, Arizona. The population is primarily white, and mostly single. At $200,200 the average home value here is a bit higher than average for the Phoenix-Mesa-Scottsdale metro area, so this probably isn't the place to look for housing bargains.5282 Zip code is located in the Mountain time zone at 33 degrees latitude (Fun Fact: this is the same latitude as Damascus, Syria!) and -112 degrees longitude.
Query: why is gibbs model of reflection good
Passage: In this reflection, I am going to use Gibbs (1988) Reflective Cycle. This model is a recognised framework for my reflection. Gibbs (1988) consists of six stages to complete one cycle which is able to improve my nursing practice continuously and learning from the experience for better practice in the future.n conclusion of my reflective assignment, I mention the model that I chose, Gibbs (1988) Reflective Cycle as my framework of my reflective. I state the reasons why I am choosing the model as well as some discussion on the important of doing reflection in nursing practice.
Query: what does a thousand pardons means
Passage: Oh, that's all right, that's all right, give us a rest; never mind about the direction, hang the direction - I beg pardon, I beg a thousand pardons, I am not well to-day; pay no attention when I soliloquize, it is an old habit, an old, bad habit, and hard to get rid of when one's digestion is all disordered with eating food that was raised forever and ever before he was born; good land! a man can't keep his functions
Query: what is a macro warning
Passage: Macro virus warning appears when no macros exist in the file in Word. When you open a Microsoft Word 2002 document or template, you may receive the following macro virus warning, even though the document or template does not contain macros: C:\¡path¿\¡file name¿contains macros. Macros may contain viruses.
[Examples End]
The query you need to process is:
Query: question
Passage:

---

**Prompt 11: Prompt of Tree Selecting**

Given a medical question and several medical decision trees, the medical question is to provide a treatment recommendation for the patient based on the provided patient information.
First, you need to analyze the relevance of each medical decision tree to the question one by one, i.e., whether the content of the decision tree aligns with the patient's condition. Then, you need to select decision trees that meet any of the following conditions:
Condition 1: The conditions of the decision tree match the patient's condition, so the outcome of the decision tree can be used for the patient's treatment.
Condition 2: The decision tree includes some information not mentioned in the patient's records, which can be used for further analysis. However, the content of the decision tree must not conflict with the patient's information.
When listing the decision trees, prioritize those more helpful in answering the question by placing them earlier.
The medical question is:
[Start of medical question]
{question}
[End of medical question]
The decision trees are:
{trees_str}
Please reply in the following format:
Analysis: In this section, analyze the relevance of each medical decision tree to the medical question one by one.
Conclusion: In this section, output the serial numbers of the selected decision trees. For example, if Decision Tree 1 and Decision Tree 3 meet the requirements, and Decision Tree 3 is more helpful, output a list: [3, 1].
Your answer is:

1512
1513
1514
1515
1516
1517
1518
1519
1520
1521
1522
1523
1524
1525
1526
1527
1528
1529
1530
1531
1532

---

**Prompt 12: Prompt of Guideline Selecting**

Please rate medical texts (on a scale of 1 to 5) according to the following rules:
5 points: It also includes the following contents:
- Compare two or more similar diseases
- Explain why this diagnosis was chosen
- Provide specific treatment methods
4 points: There are disease comparisons and treatment methods, but specific causes or drug dosages are lacking
3 points: The name and symptoms of the disease were mentioned, but no comparison was made with other possible diseases
2 points: Only stating that there is a disease without explanation, or only offering general advice (such as "Seek medical attention promptly")
1 point: No mention of any disease diagnosis-related content at all, or only health science popularization knowledge (such as disease definitions, preventive measures)
Output Format:
Only integers within the range [1-5] must be returned. Any punctuation or text is prohibited
The medical text you need to handle is:

---
