# OpenReview forum: "Medical Decision Tree-Enhanced LLMs for Interpretable Reasoning"
_ICLR.cc/2026/Conference — ICLR 2026 Conference Withdrawn Submission_

### Official Review · Reviewer_eN9X · 2025-10-29

**Soundness:** 3
**Presentation:** 3
**Contribution:** 3
**Rating:** 4
**Confidence:** 2

**Summary:**

The paper proposes a Retrieval-Augmented Generation (RAG) framework, MDT-RAG, which uses Medical Decision Trees (MDTs) as retrieval and reasoning carriers. The core workflow includes automatically extracting multi-branch MDTs from clinical guidelines (validated through structural and factual checks), summarizing MDTs into retrievable text using TreeIndex, filtering retrieved results for relevance with LLMFilter, and converting the structured MDT logic into readable IF-ELSE natural language guidance via TreeNLGen to assist LLMs in answering questions.

A large-scale MDT retrieval database was constructed (66,535 trees from ~16k guidelines), along with a medical reasoning evaluation dataset based on MIMIC-IV (2,046 entries, covering both medication recommendation and treatment planning tasks).
In comparisons across multiple models (small general-purpose models, medically specialized models, large open-source models, and commercial closed-source systems) and various RAG baselines (OQ-RAG, QR-RAG, Q2D-RAG), MDT-RAG consistently outperformed in terms of average score (AVG) and high-quality response rate (HQR), with particularly significant improvements for smaller models. Ablation studies demonstrate that TreeIndex, TreeNLGen, and LLMFilter are essential components, while sensitivity experiments on the number of trees reveal an "optimal middle ground."
The paper highlights enhanced interpretability and a step-by-step reasoning path more aligned with clinical practices.

**Strengths:**

-  The paper is well-structured, with clear figures illustrating comparisons and workflows. Formalization and implementation details are adequately provided.
-  Medical reasoning demands interpretability. MDT-RAG's structured guidance and transparent reasoning paths offer practical value for deployment. It shows significant improvements for smaller models, reducing reliance on large proprietary models.
- The experiments are solid and comprehensive.

**Weaknesses:**

- The two implementations of LLMFilter (large model judgment vs. lightweight scoring threshold) are described as "having similar effectiveness," but a quantitative comparison is lacking.
-  Additionally, the input token budget for text-based RAG and MDT-RAG is not fully equivalent (with MDT being shorter on average). Although this is disclosed in the paper and addressed through sensitivity experiments, it is recommended to provide a strict comparison under "equal token budget/equal retrieval units" conditions.

**Questions:**

Bias mitigation in LLM-as-Judge scoring: Has cross-reviewer model consistency (e.g., across different vendors/architectures) and expert human sampling validation been conducted?

Equal budget comparison: Can a strict comparison between text-based RAG and MDT-RAG under equal token budget/equal retrieval units be provided?

Quantitative comparison of the two LLMFilter implementations: What are the performance, latency, and cost statistics for each approach?
External validity: Is there an expert evaluation of case-to-path alignment or an assessment of path accuracy?

---

### Official Review · Reviewer_bRLi · 2025-10-30

**Soundness:** 1
**Presentation:** 3
**Contribution:** 2
**Rating:** 4
**Confidence:** 4

**Summary:**

The authors propose MDT-RAG, a retrieval-augmented generation framework that replaces verbose text snippets with medical decision trees (MDTs) extracted from clinical guidelines. The retrieval source comprises 15960 guideline documents, resulting in 66535 decision trees. For evaluation, the authors curate a reasoning dataset from MIMIC-IV, with around 2k patient entries and two tasks per entry (medication recommendation and treatment plan design). Experiments across five LLMs report consistent gains for MDT-RAG over text-based RAG on the proposed metrics.

**Strengths:**

- The idea of replacing unstructured text chunks with structured decision trees sounds interesting and innovative to me.
- The authors present a practical and reproducible pipeline, converting tree logic into DNF and IF–ELSE rules, and have open-sourced their implementation for transparency.
-  Across five LLMs, MDT-RAG outperforms text-RAG baselines for both medication and treatment planning tasks.
- Ablation studies are provided to justify the importance of different components

**Weaknesses:**

- The dataset construction process relies heavily on LLMs for annotation and correction. It remains unclear whether model-generated content is fully faithful to source knowledge or subject to hallucination. Although a Fact Verification step is mentioned, it is unclear which LLM is used and how its judgments align with human evaluators. This might raise concerns about the reliability of the retrieval source for MDT-RAG.
- As shown in the ablation study, MDT Filtering is an important step to ensure the high performance of MDT-RAG. However, a similar filtering mechanism can also be easily applied to text-based RAG. A fairer comparison incorporating equivalent filtering for text-RAG would clarify whether the observed gains truly stem from the MDT structure itself.
- While the authors claim DDXPlus is narrow in its evaluation scope. I don't see why it does not apply to the methods compared here. The current paper only contains evaluations on the self-constructed dataset. Evaluating on additional public datasets would verify if the performance gain is generalizable.
- All three evaluation metrics rely on LLM-based scoring, which could be subjective and biased. Analyzing the ranking quality or adding more objective metrics would strengthen the credibility of the reported results.

Minor:
Line 395 should have "Table 1 and Table 2" instead of "Table 2 and Table 2".

**Questions:**

Please see the Weaknesses.

---

### Official Review · Reviewer_WJrA · 2025-11-01

**Soundness:** 1
**Presentation:** 1
**Contribution:** 2
**Rating:** 2
**Confidence:** 3

**Summary:**

The paper presents a medical decision tree dataset and a method for medical reasoning using the medical decision trees. The medical decision tree dataset is automatically extracted from medical guidelines using LLMs. The method for medical reasoning first converts (relevant nodes in) the medical decision trees into natural language and makes diagnostic recommendations. The experimental results show some performance improvement.

**Strengths:**

1. I basically think that the dataset potentially contributes to the community (if it is publicly available).

**Weaknesses:**

1. This is just a minor point, but in Section 3.1, the paper says “we opted for multi-branch MDTs instead of conventional binary trees,” which seems weird to me. The structure of the tree seems to be determined solely based on the structure of diagnostic procedures, and it’s not our choice.
2. I cannot see how the structural correctness of the medical decision trees is evaluated.
3. As for the content correctness of medical decision trees, I’m not sure if inherent knowledge in LLMs is sufficient to evaluate it. How is the correctness guaranteed? I would like to see some discussions about this point.
4. Related to 2 and 3, it is not straightforward to see the correctness of the medical decision trees. This may depend on the paper’s position, i.e., if its focus is to construct faithful medical decision trees for potential deployment or is solely to serve as a benchmark for models. For the former, I believe expert evaluation is mandatory even for a subset of trees. For the latter, at least consistency of the trees (e.g., reachability to the final decision) should be evaluated so that the dataset can provide some ideas about the upper-bound performance.
5. It’s hard for me to see if the nodes in the proposed medical decision trees really form a tree. First of all, a node $n$ seems to contain multiple triplets, so I guess a single node $n$ forms a tree. However, then $C$ does not make much sense. Also, the logical operators in $\phi$ are associated with a single triplet, which means binary operators “AND” and “OR” are applied to a single operand? I could not understand this. This medical decision tree structure definition should be fully revised.

**Questions:**

1.	I don’t see why errors in the generated graph grow significantly when the tree is deeper. Can this be more specific?

---

### Official Review · Reviewer_S8tq · 2025-11-01

**Soundness:** 2
**Presentation:** 2
**Contribution:** 2
**Rating:** 2
**Confidence:** 4

**Summary:**

This paper introduces a method that integrates medical decision trees (MDTs) into large language models to enhance clinical reasoning. By converting guideline knowledge into structured condition–conclusion trees and rewriting them into natural language, the approach aims to make medical reasoning more interpretable and reliable.

**Strengths:**

1. The in-depth exploration of how to improve medical reasoning is meaningful and interesing, more than just finding the correct answer.
2. Using MDTs to represent clinical pathways and embedding them into the model improves interpretability and potentially enhances real-world applicability in medical contexts.

**Weaknesses:**

1. The main results rely on o4-mini’s 1–5-scale judgments on only 200 samples. This limited scope and the exclusive use of an LLM judge make the evaluation less convincing, especially without any discussion on inter-judge consistency or possible bias.

2. The entire pipeline depends heavily on large models for data construction, which could introduce systematic bias. There is also little discussion on the medical rigor of the MDT definitions and generation templates.

3. The dataset construction lacks details and transparency. It is unclear how drug and treatment recommendations were extracted and how the reference labels were linked to each case.

4. Lack of more powerful baseline. While the related work section acknowledges that structured RAG methods, such as knowledge-graph or rule-based retrieval can improve precision, the experiments only compare against text-based RAG variants (OQ/QR/Q2D) and CoT. This makes it difficult to isolate the specific advantage of using tree structures.

**Questions:**

1. Are there any objective or quantitative evaluation metrics besides LLM-based scoring?

2. How exactly was the dataset built, and how were the label references matched with the source data?

3. Beyond qualitative case studies, was there any user study or physician evaluation to measure the improvement in interpretability?

---

### Note · Authors · 2025-11-17

I have read and agree with the venue's withdrawal policy on behalf of myself and my co-authors.